# Status, reporting completeness and methodological quality of pilot randomised controlled trials in acupuncture: protocol for a systematic review

Yajun Zhang ,[1] Hantong Hu ,[2] Xiaoyu Li,[1] Jiali Lou,[1] Xiaofen He,[3] Yongliang Jiang,[3] Jianqiao Fang [3]

YZ and HH contributed equally.

YZ and HH are joint first authors.

For numbered affiliations see end of article.

**Correspondence to**
Professor Jianqiao Fang;
fangjianqiao7532@163.com

## ABSTRACT

**Introduction**  To date, there has been a lack of knowledge about the status, reporting completeness and methodological quality of pilot trials in the acupuncture field. Thus, this systematic review protocol aims to: (1) investigate publication trends and aspects of feasibility evaluated in acupuncture pilot trials; (2) identify the proportion of acupuncture pilot trials that lead to definitive trials and (3) assess the reporting completeness and methodological quality of pilot trials in acupuncture.

**Methods and analysis**  Studies of acupuncture pilot randomised controlled trials published from 2011 to 2021 will be retrieved in seven databases in January 2022, including PubMed, Web of Science, EMBASE, Cochrane Library, Chinese National Knowledge Infrastructure, Wanfang Database and Chinese Biomedical Literature Database. The methodological quality and reporting completeness of all included studies will be assessed using the risk of bias 2.0 tool (RoB 2) and the Consolidated Standards of Reporting Trials (CONSORT) extension to randomised pilot and feasibility trials, respectively. For the primary analysis, publication trends, aspects of feasibility and the proportion of pilot trials that lead to definitive trials will be analysed. A quantitative analysis of the methodological quality and reporting completeness of the included trials will be implemented by calculating the percentage of items reported in each domain of RoB 2 and CONSORT. The secondary analysis will adopt a regression analysis to identify factors associated with the reporting completeness.

**Ethics and dissemination**  Ethical approval is not required for this study. This study is planned to be submitted to a peer-reviewed academic journal.

## INTRODUCTION

Although clinical trials serve as the basis for improvement of clinical practice and establishment of evidence-based clinical guidelines, they generally consume considerable human, financial and material resources. Moreover, it is difficult to make major changes once they are conducted formally.[1]

## Strengths and limitations of this study

► This systematic review protocol will be the first to investigate the status, reporting completeness and methodological quality of pilot trials in the acupuncture field.

► The reporting completeness of pilot trials will be evaluated using the Consolidated Standards of Reporting Trials extension for Reporting Randomised Pilot and Feasibility Trials guideline.

► The methodological quality of included trials will be assessed using the Cochrane's updated risk of bias 2.0 tool.

► There may be a lack of available information to confirm whether a pilot trial has led to a definitive trial in some included studies.

► Findings from this study will be restricted to trials published between 2011 and 2021.

In such situations, if the design of the formal trial has major flaws, it will lead to a waste of resources. A previously published study[2] estimated that 85% of funds for biomedical research are wasted each year due to misdirected or poorly designed studies. Therefore, it is of significance to validate and improve scientific questions, trial designs and protocol implementations by conducting pilot trials before formal trials.[3]

Despite that several definitions of pilot trials have been put forward, most of them share the idea of conducting a study prior to a larger, more comprehensive study.[4] For example, the UK National Institute for Health Research defines a pilot trial explicitly as 'a version of the main study that is run in miniature to test whether the components of the main study can all work together', which is a widely accepted definition.[4] Therefore, whether a study can be labelled as a pilot trial

mainly depends on the primary objective of the study, which is to evaluate the feasibility and coordination of the components of an ensuing definitive trial.

Given that the design of a pilot trial is generally consistent with the subsequent formal trial, a well-designed pilot trial can not only identify potential problems that arise in a formal trial in advance but also promote the formal trial by investigating the feasibility and coordination of the study design, thereby avoiding potentially disastrous consequences if large-scale trials are conducted rashly.[5] In the guidelines of the UK Medical Research Council for designing and evaluating complex interventions, pilot studies are explicitly recommended, particularly for identifying problems that might arise in a subsequent large-scale trial of a complex intervention.[6 7]

Acupuncture belongs to complex interventions, so it is more necessary to conduct pilot trials before a formal acupuncture trial. In recent years, the benefits of pilot trials have been verified in a number of high-quality clinical trials of acupuncture. For example, based on results obtained from a series of pilot trials involving acupuncture for treating osteoarthritis,[8–10] Liu *et al* not only fully examined the study procedure to determine the feasibility of study designs, but also obtained important pilot data, such as optimal acupuncture treatment frequency, patient compliance rate, and primary outcome measures. Subsequently, their research team was able to improve the study design in the large-scale definitive trial and implemented it successfully, which was published in the high-impact journal *Arthritis & Rheumatology*.[11]

Despite notable value and impacts, the role of pilot studies remains underestimated in scientific research. Moreover, previous studies demonstrated that only a very small percentage of pilot trials lead to definitive trials.[12] To the best of our knowledge, there have been no published studies that aimed to investigate the status and quality of pilot trials in the acupuncture field. Key questions concerning the publication trend, impact, feasibility, reporting completeness and methodological quality of acupuncture pilot trials are urgent to be solved. Thus, this systematic review protocol is designed and it aims to: (1) investigate the overall publication trends and the aspects of feasibility evaluated in acupuncture pilot trials; (2) identify the proportion of acupuncture pilot trials that lead to definitive trials and (3) assess the reporting completeness and methodological quality of pilot trials in the acupuncture field.

## METHODS
This systematic review protocol is drafted according to the Preferred Reporting Items for Systematic Reviews and Meta-Analyses for Protocols guidelines.

### Eligibility criteria
A previously published tutorial[5] regards that 'pilot' trials belong to the dimension of 'feasibility' trials and a pilot trial is synonymous with a feasibility study that aims to guide the planning of a large-scale investigation. Based on the main goals of this review and references of similar studies,[13–15] the boundary between pilot and feasibility trials will not be imposed in our study. We use the term 'pilot' trial to refer to studies that will be reviewed here and the term 'feasibility' if the original authors of the included publications adopted this term rather than 'pilot'. Therefore, published trials will be eligible if studies were defined and reported explicitly as pilot trials within the published paper itself by the original authors. Additionally, as has been done previously in similar studies,[13–15] publications labelled as 'feasibility', 'preliminary', 'exploratory', 'proof-of-concept' and 'vanguard' studies/trials' will also be considered for eligibility because these terms are often used interchangeably with the term 'pilot'.[12 16 17] In addition, given that the International Committee of Medical Journal Editors proposed that all trials conducted after July 2005 should be registered in a public registry platform[18 19] and the latest update of the Consolidated Standards of Reporting Trials (CONSORT) statement was released in 2010,[20] only publications of pilot trials published after 2010 will be evaluated for eligibility in our review.

Studies reported as acupuncture pilot or feasibility trials will be eligible for this review if they meet the following criteria: (1) included the evaluation of at least one kind of acupuncture intervention; (2) randomised control trials (RCTs) conducted in humans and (3) trials published from 2011 to 2021 in any language. The term 'acupuncture' in this review is defined as interventions involving penetration of specific points on the skin by needling, regardless of needling and stimulation modalities, such as manual acupuncture, electroacupuncture, auricular acupuncture and warm needling, or a combination of these modalities. Studies will be excluded if: (1) they are non-RCTs or there are no control groups, such as trials with a pre–post design or an observational study design and (2) no acupuncture intervention was used solely or in combination in the trial.

### Databases and search strategy
To search published studies of acupuncture pilot trials, we will conduct a comprehensive retrieval on four English databases (PubMed, Web of Science, EMBASE, Cochrane Library) and three Chinese databases (Chinese National Knowledge Infrastructure, Wanfang Database, Chinese Biomedical Literature Database) in January 2022. The time range of published studies will be limited from 2011 to 2021.

Search strategies will consist of two components: interventions (manual acupuncture, electroacupuncture, etc) and study type (pilot trial). Search strategies will combine subject headings (eg, Medical Subject Headings (MeSH) for PubMed) and keywords (acupuncture, eletroacupuncture, pilot, feasibility, preliminary, exploratory, proof-of-concept, vanguard, etc). The PubMed retrieval strategy will be adapted for other databases by replacing MeSH terms with proper subject headings (if applicable) and

| No. | Search items |
|---|---|
| #1 | Randomized controlled trial [pt] |
| #2 | Controlled clinical trial [pt] |
| #3 | Randomized OR Randomised(Title/Abstract) |
| #4 | Clinical trials [MeSH] |
| #5 | Randomly(Title/Abstract) |
| #6 | Trial(Title/Abstract) |
| #7 | #1 OR #2 OR #3 OR #4 OR #5 OR #6 |
| #8 | Pilot Projects OR Feasibility Studies OR Proof of Concept Study(MeSH) |
| #9 | pilot OR feasibility OR preliminary OR exploratory OR proof-of-concept OR vanguard(Title/Abstract) |
| #10 | #8 OR #9 |
| #11 | Acupuncture Therapy [MeSH] |
| #12 | (acupuncture OR electroacupuncture OR electro-acupuncture OR manual acupuncture OR auricular acupuncture OR ear acupuncture OR warm needling OR warming-needle moxibustion)(Title/Abstract) |
| #13 | #11 OR #12 |
| #14 | #7 AND #10 AND #13 |

**Table 1** Search strategy in PubMed

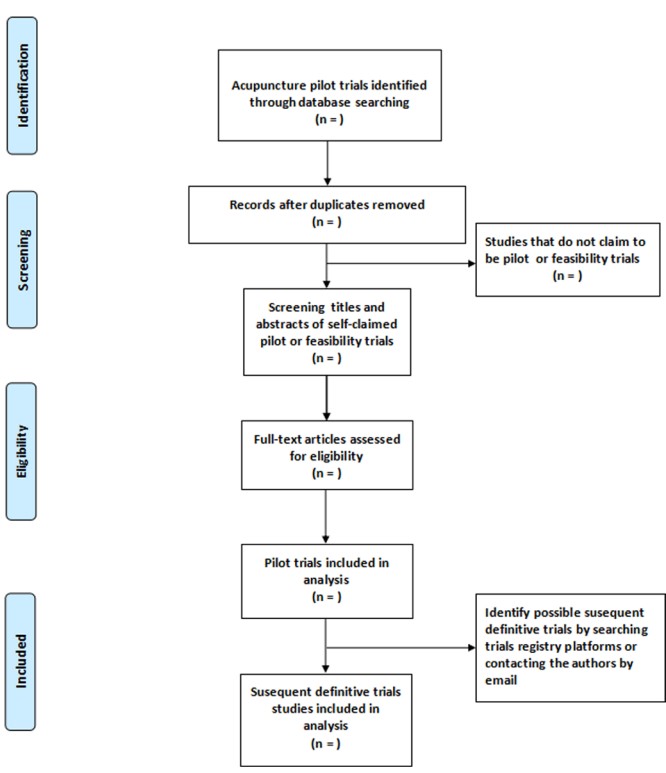

**Figure 1** The flow diagram of study search and screening.

maintaining the same keywords. The retrieval strategy for PubMed is presented in table 1.

## Screening and study selection

All retrieval records identified from seven electronic databases will be exported into Mendeley Reference Manager software (Elsevier Publishing, 2013) for screening. Following the removal of duplicate retrieval literature using Mendeley, two investigators (YZ and HH) will scan titles and abstracts of retrieval studies independently to determine studies that claim to be acupuncture pilot or feasibility trials. In the first round of screening, two independent investigators will browse the titles and abstracts of publications to confirm eligibility. In the second round of screening, the full text of potentially qualified studies will be downloaded to determine eligibility based on the inclusion criteria and exclusion criteria. A third senior investigator (J-QF) will be invited to solve any disagreements between the two investigators during study selection. Additionally, if the number of all excluded studies is ≤500, a description of the specific reasons for each excluded study will be provided in an Excel sheet. If the total number is more than 500, the reference lists of excluded studies will be supplied in an online appendix. The flowchart of study search and screening is presented in figure 1.

## Data extractions

A predesigned electronic data extraction form using Microsoft Excel (Microsoft, 2018) will be developed for this study based on the recommendation from the

CONSORT group.[20] Two investigators (YZ and HH) will extract data of all included trials independently using the data extraction form, which consists of various aspects such as (1) the publication title; (2) the name of the first author and corresponding author; (3) the number of authors; (4) the language of publication; (5) the country that conducts the trial; (6) the publication year; (7) study designs; (8) conditions being treating; (9) types of intervention; (10) types of controls; (11) trial registration platforms (if applicable); (12) the impact factor of the journal (if applicable) in the publication year based on the Journal Citation Reports (https://jcrincitesthomson-reuters. com); (13) sample sizes; (14) sources of funding; (15) outcome measures; and so on.

With the attempt to improve consistency and accuracy between the investigators, prior to full-scale data extraction, the feasibility of data extraction forms and between-reviewer agreements will be investigated via kappa coefficients by choosing 10 randomly selected studies. If inadequate consistency and accuracy are identified, extra training on data extraction will be conducted. A third senior reviewer (J-QF) will be invited to verify the accuracy of all extracted data and settle any discrepancies between the two investigators.

## Assessment of the aspects of feasibility
### Rationales for conducting a pilot trial

We will classify whether the original authors of the pilot study stated the rationales for conducting a pilot trial (1 for yes; 0 for no). If yes, we will further classify the rationales for conducting pilot and feasibility studies based on a widely acknowledged tutorial, which suggests 4 broad

classifications to sort out the rationales for performing pilot trials as follows.[5]

1. Process: This involves the assessment of the feasibility of the steps that need to take place as part of the future definitive trial, such as estimation of recruitment rates, retention rates, etc.
2. Resources: Concerning the evaluation of budget and time problems that might arise during the future definitive trial.
3. Management: This involves the evaluation of potential human and data optimisation problems, such as personnel and data management issues in different research centres.
4. Scientific: This involves the evaluation of treatment safety, determination of treatment dose and estimation of treatment effect size, such as the acquisition of preliminary data for sample size estimation in the future definitive trial

If there is more than one classification identified for some included studies, all applicable classifications will be coded.

### Conclusions of a pilot trial

In general, the conclusions of a pilot or feasibility trial can be grouped under several classifications as follows[15]: (1) the study protocol is not feasible and the future full-scale definitive trial is not recommended to be conducted; (2) the study protocol is feasible to a certain extent but needs to be modified before proceeding to a future full-scale definitive trial; (3) the study protocol does not need to be modified, but the future full-scale definitive trial needs to be performed under close monitoring and (4) the study protocol is completely feasible. We will classify the conclusion of each included pilot trial according to the above-mentioned classifications.

### Whether a future definitive trial is conducted

The strategies for identifying potential definitive trials as far as possible in a similar study[13] will be used in our systematic review. In detail, with all the included pilot trials determined, a secondary search will be performed in the same databases to identify corresponding definitive trials. In addition, the International Clinical Trials Registry Platform, Clinicaltrials Registry (https://clinicaltrials.gov/) and Chinese Clinical Trial Registry (http://www.chictr.org.cn) will be searched to ascertain whether a definitive trial was conducted (or in progress). The search terms for definitive trials in databases and trials registry platforms are based on some unique information of the published pilot studies (eg, the name of the corresponding authors, the name of one or more coauthors, the major part of the article title). Finally, if no definitive trials are identified based on these retrieval methods, we will try to contact the corresponding authors of the published pilot study by email and query whether a subsequent definitive trial was in progress, conducted, or submitted for publication.

With the obtained results, we will classify whether a definitive trial was conducted following the pilot trial (1 for yes; 0 for no) and calculate the proportion of pilot trials that lead to definitive trials. Meanwhile, we will investigate whether the pilot data were used for sample size calculations in the subsequently published definitive trial (1 for yes; 0 for no).

### Reporting completeness assessment of pilot trials

The reporting completeness of each included study will be evaluated in accordance with the CONSORT extension for reporting randomised pilot and feasibility trials, which consist of a 40-item checklist.[21] The CONSORT checklist mainly consists of multiple items regarding titles, abstracts, objectives, trial designs, methods, results, conclusions, limitations, registrations and fundings.[22] In our study, each item of the CONSORT checklist will be scored as 'reported', 'inadequately reported', 'not reported' or 'not applicable', as has been done previously in a similar study in other research filed.[14] Reporting completeness assessment will be performed by two independent investigators (XL and JL). Any disagreement between the two reviewers will be solved through discussion or be arbitrated by a third senior reviewer (YJ) until a consensus is reached.

### Methodological quality assessment of pilot trials

The methodological quality of the included studies will be assessed by two independent reviewers (XL and JL) using the Cochrane collaboration's updated risk of bias 2.0 (RoB 2.0) tool, in which the ROB of each included trial will be assessed based on the following domains: (1) randomisation process; (2) deviations from intended interventions; (3) missing outcome data; (4) measurement of the outcome and (5) selection of the reported result. Each domain will be rated as 'low', 'high' or 'some concerns'. The overall ROB for a single trial will be also classified as 'low' (ROB is low for all domains), 'some concerns' (some concerns in at least one domain) and 'high' (high ROB for at least one domain or some concerns for multiple domains). Disagreements between the two independent reviewers will be judged by the third senior reviewer (J-QF).

### Statistical analysis

For the primary analysis, the number of publications per year will be calculated and presented in a tendency chart to analyse overall publication trends. The aspects of feasibility assessed in the included pilot trials will be tabulated. The proportion of pilot trials that lead to definitive trials will be calculated. A quantitative analysis of the reporting completeness and methodological quality of all included acupuncture pilot trials will be conducted. In detail, the frequency that each item is scored as 'reported', 'inadequately reported', 'not reported' and 'not applicable' for the CONSORT checklist will be tabulated; the summary score will be calculated as mean±SD. The frequency that each domain is rated as 'low', 'high' or 'some concerns' for ROB 2.0 will be tabulated.

For the secondary analysis, regression analysis will be applied to determine which study characteristic (eg, language of publication, the number of authors, trial registration or not, trial funded or not) has associations with the summary score for the CONSORT checklist. A p value less than 0.05 is considered significant. All analyses will be performed using SPSS V.25.0 software (IBM).

## Patient and public involvement

Patients and the public will not involve in this study.

## Ethics and dissemination

Ethical approval is not required for this study because no personal and private information of individuals will be involved. This study is planned to be submitted to a peer-reviewed academic medical journal.

## DISCUSSION

The pilot trial is a key component of clinical research to obtain important preliminary data and information for improving the study design and methodological quality of future definitive trials.[5] Based on a brief review of acupuncture studies published in journals with high impact factors in recent years, the majority of them are conducted on the basis of earlier published pilot trials. For example, Xu *et al* conducted an acupuncture pilot trial[23] and found that electroacupuncture at Zhongliao (BL33) and Huiyang (BL35) could lead to less urine leakage in women of stress urinary incontinence with an acceptable safety and feasibility profile. This pilot trial[23] provided an adequate basis for the acupoint selection and feasibility for a subsequent multicentre RCT published in *JAMA*.[24] Additionally, Xue *et al* and Wu *et al* performed a series of pilot trials investigating the effect of acupuncture on functional constipation,[25–27] which serve as a solid basis for the subsequent definitive trial of acupuncture for chronic severe functional constipation published in *Annals of Internal Medicine*.[28] Such situations indicate that acupuncture researchers are paying increasing attention to the value of pilot trials.

Moreover, acupuncture belongs to complex treatment interventions and it is a key component of complementary and complementary medicine. Due to the unique nature of acupuncture operations, some patients with specific diseases may be unwilling or unable to receive acupuncture treatment, especially in western countries. Thus, before the formal trial, preliminary data are very valuable. For example, it is crucial to obtain preliminary data regarding the recruitment rate, drop-out rate, and treatment frequency prior to a subsequent definitive trial. The interpretation of these preliminary results will directly determine whether a future full-scale formal trial should be conducted, or whether the current study protocol needs improvement. It can also provide pilot data for sample size calculations in the subsequent definitive trial. In such a scenario, pilot trials play a crucial role in ensuring a feasible and sound methodological approach in the future definitive trial.

To date, however, there has been a great lack of knowledge about the publication trend, reporting completeness, and methodological quality of acupuncture pilot trials. To the best of our knowledge, there have been no previously published similar studies in the acupuncture field. This will be the first systematic review protocol that aims mainly to investigate the status, reporting completeness, and methodological quality of pilot RCTs in the acupuncture field. Therefore, this study is of significance, which will guide acupuncture clinical research for both researchers and clinicians in the future.

It is worthnoting that our study has several strengths. The first strength is to conduct a broad retrieval to identify all eligible acupuncture pilot trials published from 2011 to 2021 in seven mainstream electronic databases. Subsequently, the status and overall publication trends will be revealed. Second, this systematic review will assess the reporting completeness of each included pilot RCT following the 40-item CONSORT extension for reporting randomised pilot and feasibility trials,[21] which is a highly acknowledged and recommended tool.[29] Transparent and complete reporting is a key principle of rigorous research, and reporting tools (eg, CONSORT) enable both authors and readers to interpret studies. Specifically, the CONSORT extension for reporting randomised pilot and feasibility trials[21] is developed to provide reporting guidance for any randomised study in which a future definitive RCT, or part of it, is conducted on a smaller scale, regardless of its design (eg, cluster, crossover) or the terms used by authors to label the study (eg, pilot, feasibility). A previous systematic review indicated that the application of the CONSORT checklist has close associations with improvement in the proper reporting of RCTs.[30] Thus, findings of our systematic review will provide a better understanding of the reporting completeness and transparency of current studies. On the basis, it will further shed the light to promote the appropriate reporting of acupuncture pilot trials in the future. Third, the methodological quality of each included pilot RCT will be assessed using Cochrane's updated ROB 2.0 tool,[31] which is a well-established and reliable method. A previous study[32] has proved its fair reliability and good construct validity and ROB 2.0 is also commonly used in systematic reviews with similar research goals.[33 34]

Nevertheless, several limitations of this study should be addressed. First, despite that multiple measures will be adopted to identify subsequent definitive trials on the basis of the included pilot studies, there may be a lack of available information to confirm whether an acupuncture pilot trial has led to a definitive trial in some included studies. Second, given that the CONSORT extension for reporting randomised pilot and feasibility trials is applied to evaluate the reporting completeness and transparency of included studies and the latest update of the CONSORT statement was released in 2010,[20] only acupuncture pilot RCTs published from 2011 to 2021 will be included for analysis in our review. Third, it is difficult to clearly distinguish between the small studies and pilot studies in some situations. Some clinical studies are conducted based on previous small studies, but the terms 'pilot' or 'feasibility' probably don't appear in these

published small studies. It could affect the comprehensiveness of literature retrieval.

**Author affiliations**
[1]The Third Clinical Medical College, Zhejiang Chinese Medical University, Hangzhou, China
[2]Department of Acupuncture and Moxibustion, The Third Affiliated Hospital of Zhejiang Chinese Medical University, Hangzhou, China
[3]Key Laboratory of Acupuncture and Neurology of Zhejiang Province, Department of Neurobiology and Acupuncture Research, The Third Clinical Medical College, Zhejiang Chinese Medical University, Hangzhou, China

**Contributors** HH and J-QF were responsible for the conception and design of the protocol. XH and YJ provided critical insights into the methodology. YZ drafted the first version of the manuscript. JL and XL revised the manuscript. All authors have critically reviewed and approved the final manuscript.

**Funding** The trial is financially supported by the National Key R&D Programme of China (NO.2018YFC1704600).

**Competing interests** None declared.

**Patient and public involvement** Patients and/or the public were not involved in the design, or conduct, or reporting, or dissemination plans of this research.

**Patient consent for publication** Not applicable.

**Provenance and peer review** Not commissioned; externally peer reviewed.

**ORCID iDs**
Yajun Zhang http://orcid.org/0000-0003-0153-5878
Hantong Hu http://orcid.org/0000-0001-8759-5083
Jianqiao Fang http://orcid.org/0000-0003-4499-0352

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
