## [Reviewer comments · BMJ Open]

ARTICLE DETAILS

TITLE (PROVISIONAL)	The status, reporting completeness and methodological quality of pilot randomized controlled trials in acupuncture: protocol for a systematic review
AUTHORS	Zhang, Yajun; Hu, Hantong; Li, Xiaoyu; Lou, Jiali; He, Xiaofen; Jiang, Yongliang; Fang, Jian-Qiao

VERSION 1 – REVIEW

REVIEWER	Liu, Cun-Zhi Dongfang Hospital, Beijing University of Chinese Medicine, Department of Acupuncture and Moxibustion
REVIEW RETURNED	25-Aug-2021

GENERAL COMMENTS	This is a well written, comprehensive protocol to investigate status and reporting quality of acupuncture pilot randomized controlled trials. 1. How to determine the relationship between pilot and definitive studies is the difficulty of this research, the authors need to formulate detailed policies and clarify them in the protocol.2. It is recommended to use the Cochrane collaboration's tool and jadam scale to assessing risk of bias on the quality evaluation.3. How to distinguish between the small studies and pilot studies? In fact, many of clinical studies were refinement based on previous small studies, but the word "pilot" or "feasibility" probably didn't come up in these studies. It could affect the comprehensiveness of retrieval.
---

REVIEWER	Guo, Y Tianjin University of Traditional Chinese Medicine
REVIEW RETURNED	24-Sep-2021

GENERAL COMMENTS	1. This study is the first to evaluate the methodological quality of pilot trials in the field of acupuncture. This work is of significance for a better understanding of the methodological quality of current studies and improve the design and reporting quality of acupuncture pilot trials.2. Please supplement the registration number.3. Please describe in detail how the author handles a large number of search results and various data sources in many different languages, and whether uses relevant software and tools.4. Whether appropriate reasons for each excluded study will be provided by the investigators?5. The pre-designed electronic data extraction form should contain richer contents, such as the title, first author's name, outcome index and the other information, so as to tracking the research information,
---

	to make clear whether the pilot trials lead to definitive trials and analysis the feasibility of some elements in the study design. 6.The study protocol did not limit the intervention methods, intervention measures of the control group and observation indicators. When comparing different groups, the operation of the intervention group and control group as well as the outcomes contain a large number of mixed factors. Although this is not the main goal of the study, it will affect the evaluation of research quality and results, please explain in detail how the researcher will integrate these data. 7. There are some controversies in #8 Humans[MeSH] in the retrieval strategy shown in Table 1. Many of the clinical studies with humans as the research object do not mention “humans”, but refer to “subjects”, “participants” or patients with specific diseases,So taking humans[MeSH] as retrieval word may leave out some studies. 8.Please further enrich the discussion. 9.Please pay attention to the format of references, such as references [10] and [11], the name of the journal is missing.
--	---

VERSION 1 – AUTHOR RESPONSE

Reviewer: 1

Dr. Cun-Zhi Liu, Dongfang Hospital, Beijing University of Chinese Medicine

Comments to the Author:

1. This is a well written, comprehensive protocol to investigate status and reporting quality of acupuncture pilot randomized controlled trials.

Response:

Dear reviewer, we appreciate your positive comments on our manuscript. Moreover, according to your very constructive and helpful suggestions, we have revised our manuscript very carefully. All changes have been highlighted in red color in the revised manuscript. Please check it for verification.

2. How to determine the relationship between pilot and definitive studies is the difficulty of this research, the authors need to formulate detailed policies and clarify them in the protocol.

Response:

Thank you for your constructive comments. We have added detailed policies to determine the relationship between pilot and definitive studies in the revised manuscript. Moreover, we also provide detailed strategies for identifying potential definitive trials in the revised manuscript. It's worth noting that such policies and strategies have been used in previous studies with similar goals in other research field[13]. Nevertheless, despite that multiple policies and strategies will be adopted in our study, we agree with your opinion that the determination of the relationship between pilot and definitive studies may be the difficulty of our research, so we also address it as one of the limitations of our study in the 'Discussion' section. For your earliest convenience, you could also read it here as follows:

(Line 20-28, Page 9 and Line 1-5, Page 10): 'The strategies for identifying potential definitive trials as far as possible in a similar study [13] will be utilized in our systematic review. In detail, with all the included pilot trials determined, a secondary search will be performed in the same databases to identify corresponding definitive trials. In addition, the International Clinical Trials Registry Platform (ICTRP), Clinicaltrials Registry (<https://clinicaltrials.gov/>), and Chinese Clinical Trial Registry (<http://www.chictr.org.cn>) will be searched to ascertain whether a definitive trial was conducted (or in progress). The search terms for definitive trials in databases and trials registry platforms are based on some unique information of the published pilot studies (e.g. the name of the corresponding authors, the name of one or more co-authors, the major part of the article title). Finally, if no definitive trials are identified based on these retrieval methods, we will try to contact the corresponding authors of the published pilot study by email and query whether a subsequent definitive trial was in progress,

conducted, or submitted for publication’.

(Line 8-12, Page 14): ‘Limitations: First, despite that multiple measures will be adopted to identify subsequent definitive trials on the basis of the included pilot studies, there may be a lack of available information to confirm whether an acupuncture pilot trial has led to a definitive trial in some included studies’.

References

[13] Desai B, Desai V, Shah S, et al. Pilot randomized controlled trials in the orthopaedic surgery literature: a systematic review. *BMC Musculoskelet Disord.* 2018;19(1):412.

3. It is recommended to use the Cochrane collaboration’s tool and Jadad scale to assessing risk of bias on the quality evaluation.

***Comment from the Editor: One commonly-used scale was developed by Jadad and colleagues for randomized trials in pain research (Jadad 1996). The use of this scale is explicitly discouraged. As well as suffering from the generic problems of scales, it has a strong emphasis on reporting rather than conduct, and does not cover one of the most important potential biases in randomized trials, namely allocation concealment

(see [https://handbook-5-](https://handbook-5-1.cochrane.org/chapter_8/8_3_3_quality_scales_and_cochrane_reviews.htm)

[1.cochrane.org/chapter_8/8_3_3_quality_scales_and_cochrane_reviews.htm](https://handbook-5-1.cochrane.org/chapter_8/8_3_3_quality_scales_and_cochrane_reviews.htm)).

Response:

Thank you so much for your constructive suggestion and we agree with it. Accordingly, we will adopt the Cochrane collaboration’s updated risk of bias 2.0 (RoB 2.0) tool for assessing risk of bias on the quality evaluation in the revised manuscript (Line 25-28, Page 10 and Line 1-8, Page 11). We don’t prefer to use the Jadad scale because this tool is not widely accepted, as pointed out by the editor. For your earliest convenience, you could also read it here as follows:

‘2.8 Methodological quality assessment of pilot trials The methodological quality of the included studies will be assessed by two independent reviewers (XYL and JLL) using the Cochrane collaboration’s updated risk of bias 2.0 (RoB 2.0) tool, in which the ROB of each included trial will be assessed based on the following domains: 1) Randomization process; 2) deviations from intended interventions; 3) missing outcome data; 4) measurement of the outcome; and 5) selection of the reported result. Each domain will be rated as ‘low’, ‘high’, or ‘some concerns’. The overall ROB for a single trial will be also classified as ‘low’ (ROB is low for all domains), ‘some concerns’ (some concerns in at least one domain), and ‘high’ (high ROB for at least one domain or some concerns for multiple domains). Disagreements between the two independent reviewers will be judged by the third senior reviewer (JQF).’

4. How to distinguish between the small studies and pilot studies? In fact, many of clinical studies were refinement based on previous small studies, but the word “pilot” or “feasibility” probably didn’t come up in these studies. It could affect the comprehensiveness of retrieval.

Response:

Thank you for your helpful comments. As has been done previously in similar studies in other research field [13-15], publications labeled as ‘pilot’, ‘feasibility’, ‘preliminary’, ‘exploratory’, ‘proof-of-concept’, and ‘vanguard’ studies/trials’ will be screened for eligibility in our review. But we agree with your opinion that in some situations it is difficult to distinguish between the small studies and pilot studies, so we also address it as one of the limitations of our study in the ‘Discussion’ section of the revised manuscript (Line 16-20, Page 14). For your earliest convenience, you could also read it here as follows:

‘Limitations..... Third, it is difficult to clearly distinguish between the small studies and pilot studies in some situations. Some clinical studies are conducted based on previous small studies, but the terms ‘pilot’ or ‘feasibility’ probably don’t appear in these published small studies. It could affect the comprehensiveness of literature retrieval’

References

- [13] Desai B, Desai V, Shah S, et al. Pilot randomized controlled trials in the orthopaedic surgery literature: a systematic review. *BMC Musculoskelet Disord*. 2018;19(1):412.
- [14] Scola LFC, Moseley AM, Thabane L, et al. A methodological survey on reporting of pilot and feasibility trials for physiotherapy interventions: a study protocol. *BMJ Open*. 2019;9(5):e020580.
- [15] Kaur N, Figueiredo S, Bouchard V, et al. Where have all the pilot studies gone? A follow-up on 30 years of pilot studies in *Clinical Rehabilitation*. *Clin Rehabil*. 2017;31(9):1238-1248.
-

Reviewer: 2

Dr. Y Guo, Tianjin University of Traditional Chinese Medicine

Comments to the Author:

1. This study is the first to evaluate the methodological quality of pilot trials in the field of acupuncture. This work is of significance for a better understanding of the methodological quality of current studies and improve the design and reporting quality of acupuncture pilot trials.

Response:

Dear reviewer, we appreciate your positive comments on the significance of our study. Moreover, according to your very constructive and helpful suggestions, we have revised our manuscript very carefully. All changes have been highlighted in red color in the revised manuscript. Please check it for verification.

2. Please supplement the registration number.

Response:

Thank you for your helpful suggestion. We have provided the registration information in the revised manuscript (Line 14-16, Page 5). For your earliest convenience, you could also read it here as follows:

'This study protocol has been prospectively registered on 12 April 2021 in the Open Science Framework with a unique identification link (<https://osf.io/9kmb3/>).'

It is worth noting that the unique identification link granted by 'Open Science Framework' is similar to the registration number granted by another registry platform 'PROSPERO'.

3. Please describe in detail how the author handles a large number of search results and various data sources in many different languages, and whether uses relevant software and tools.

Response:

We have added such information in the section '2.4 Screening and study selection' of the revised manuscript (Line 9-11, Page 7) according to your suggestions. For your earliest convenience, you could also read it here as follows:

'All retrieval records identified from 7 electronic databases will be exported into Mendeley Reference Manager software (Elsevier Publishing, 2013) for screening. Following the removal of duplicate retrieval literature using Mendeley, two investigators (YJZ and HTH) will.....'

4. Whether appropriate reasons for each excluded study will be provided by the investigators?

Response:

Thank you for your query. We have added relevant information in the revised manuscript (Line 19-22, Page 9). For your earliest convenience, you could also read it here as follows:

'2.4 Screening and study selection Additionally, if the number of all excluded studies is ≤ 500 , a description of the specific reasons for each excluded study will be provided in an Microsoft Excel sheet. If the total number is more than 500, the reference lists of excluded studies will be supplied in an online appendix'

5. The pre-designed electronic data extraction form should contain richer contents, such as the title, first author's name, outcome index and the other information, so as to tracking the research information, to make clear whether the pilot trials lead to definitive trials and analysis the feasibility of

some elements in the study design.

Response:

Thank you for your suggestion. We have updated the electronic data extraction form with richer contents, such as the publication title, first author's name, corresponding author's name, outcome measures and other information. We have revised it in the revised manuscript (Line 2-9, Page 8). Please check it for verification.

6. The study protocol did not limit the intervention methods, intervention measures of the control group and observation indicators. When comparing different groups, the operation of the intervention group and control group as well as the outcomes contain a large number of mixed factors. Although this is not the main goal of the study, it will affect the evaluation of research quality and results, please explain in detail how the researcher will integrate these data.

Response:

Thank you for pointing out this aspect. Based on your comment as well as suggestions from the other reviewer, we have added the methodological quality assessment of the included studies in our revised manuscript, which is evaluated by the Cochrane collaboration's updated risk of bias 2.0 (RoB 2.0) tool. According to your suggestions, we have addressed these contents in the 'Method' section of the revised manuscript (Line 25-28, Page 10 and Line 1-8, Page 11). For your earliest convenience, you could also read it here as follows:

'2.8 Methodological quality assessment of pilot trials The methodological quality of the included studies will be assessed by two independent reviewers (XYL and JLL) using the Cochrane collaboration's updated risk of bias 2.0 (RoB 2.0) tool, in which the ROB of each included trial will be assessed based on the following domains: 1) Randomization process; 2) deviations from intended interventions; 3) missing outcome data; 4) measurement of the outcome; and 5) selection of the reported result. Each domain will be rated as 'low', 'high', or 'some concerns'. The overall ROB for a single trial will be also classified as 'low' (ROB is low for all domains), 'some concerns' (some concerns in at least one domain), and 'high' (high ROB for at least one domain or some concerns for multiple domains). Disagreements between the two independent reviewers will be judged by the third senior reviewer (JQF)'

7. There are some controversies in #8 Humans[MeSH] in the retrieval strategy shown in Table 1. Many of the clinical studies with humans as the research object do not mention "humans", but refer to "subjects" , "participants" or patients with specific diseases,So taking humans[MeSH] as retrieval word may leave out some studies.

Response:

Thank you for your helpful suggestions. We have canceled 'humans [MeSH]' as the retrieval term, in case we may leave out some eligible studies. Please see the updated Table 1 for a detailed retrieval strategy in the revised manuscript (Page 21).

8. Please further enrich the discussion.

Response:

Thank you for your suggestions. We enrich the content of the 'Discussion' section in the revised manuscript (Page 12-14) according to your suggestions. For example, we added the discussion regarding the reliability of the tool for assessing the methodological quality of the included studies (Page 14). We discussed more contents regarding the value and importance of acupuncture pilot trials (Page 12). We also address more limitations of our study in the 'Discussion' section (Page 14).

9. Please pay attention to the format of references, such as references [10] and [11], the name of the journal is missing.

Response:

Sorry for our negligence and mistakes. We have revised the format of references by providing full information in the revised manuscript (Page 15-18).

VERSION 2 – REVIEW

REVIEWER	Liu, Cun-Zhi Dongfang Hospital, Beijing University of Chinese Medicine, Department of Acupuncture and Moxibustion
REVIEW RETURNED	21-Oct-2021
GENERAL COMMENTS	The authors have adequately answered to the comments.